# Risk Factors of Suicidal Ideation in Iraqi Crystal Methamphetamine Users

**DOI:** 10.3390/brainsci13091279

**Published:** 2023-09-02

**Authors:** Ahmed Al-Imam, Marek A. Motyka, Beata Hoffmann, Hussein Al-Ka’aby, Manal Younus, Nesif Al-Hemiary, Michal Michalak

**Affiliations:** 1Department of Computer Science and Statistics, Doctoral School, Poznan University of Medical Sciences, 61-806 Poznan, Poland; 2Department of Anatomy and Cellular Biology, College of Medicine, University of Baghdad, Baghdad 10047, Iraq; 3Barts and the London School of Medicine and Dentistry, Queen Mary University of London, London E1 2AD, UK; 4Institute of Sociological Sciences, University of Rzeszow, 35-959 Rzeszów, Poland; mmotyka@ur.edu.pl; 5Institute of Applied Social Sciences, University of Warsaw, 00-927 Warsaw, Poland; beata.hoffmann@uw.edu.pl; 6Department of Psychiatry, Baghdad Medical City Teaching Complex, Baghdad 10047, Iraq; hussainalkabby@gmail.com (H.A.-K.); nesif.hemiary@comed.uobaghdad.edu.iq (N.A.-H.); 7Iraqi Pharmacovigilance Centre, Ministry of Health, Baghdad 10047, Iraq; iraqiphvc@moh.gov.iq; 8The Middle East Chapter, The International Society of Pharmacovigilance (ISoP), London SW12 0HS, UK; 9Council for International Organizations of Medical Sciences (CIOMS), 1218 Geneva, Switzerland; 10Department of Psychiatry, College of Medicine, University of Baghdad, Baghdad 10047, Iraq; 11Psychiatry Council, The Iraqi Board for Medical Specializations, Baghdad 10047, Iraq; 12Department of Computer Science and Statistics, Poznan University of Medical Sciences, 61-806 Poznan, Poland; michal@ump.edu.pl

**Keywords:** addiction studies, causality assessment and causal inference, Middle East, psychiatric and behavioral problems, psychoactive substances, psychostimulants, quasi-psychedelics, suicide and parasuicide

## Abstract

Crystal methamphetamine, a potent psychostimulant, presents a complex clinical landscape. However, insights into the predisposing factors for suicidal tendencies among Iraqi users remain limited. Our study delves into these risks among 165 patients. Rigorous multivariable analysis was conducted, employing binary logistic regression, drawing from patients from Baghdad Medical City and Ibn-Rushd Teaching Hospital. Most participants were in their third decade (26.62 ± 0.53 years). Regarding suicidal ideation, our model demonstrated robust accuracy, supported by the Hosmer–Lemeshow goodness-of-fit test (NagelKerke’s R^2^ = 0.49, accuracy = 79.4%, *p* = 0.885). Notably, chronic methamphetamine use exceeding a year (OR = 6.15, *p* = 0.001), absence of psychological trauma (OR = 4.58, *p* = 0.006), and incidence of visual hallucinations (OR = 4.52, *p* = 0.001) rendered users more susceptible to suicidal ideation. Furthermore, our investigation unveiled risk factors tied to psychotic features and withdrawal manifestations. The study underscores pivotal predictors of suicidal ideation, warranting interdisciplinary vigilance from psychiatrists, clinical psychologists, and social workers. By monitoring at-risk individuals, progression toward the intricate spectrum of suicide can be potentially averted. These findings illuminate the urgency of tailored interventions for crystal methamphetamine users, contributing to enhanced holistic care.

## 1. Introduction

Amphetamine and its derivatives, collectively referred to as substituted amphetamines, are potent psychostimulants [1,2]. Amphetamines fall into the category of sympathomimetics and analeptics. According to Chachan and Al-Hemiary (2022), the primary amphetamines available in Iraq are dextroamphetamine and methamphetamine [3]. On the street, amphetamines are known by various names, including ice, crystal, crystal meth, and crystalline meth [3,4]. In the United States, commonly encountered amphetamines encompass dextroamphetamine (Dexedrine), methamphetamine (Desoxyn), a mixed dextroamphetamine–amphetamine salt (Adderall), and the amphetamine-like compound methylphenidate (Ritalin) [3,4]. 

As reported in the World Drug Report 2022 (WDR 2022), amphetamine-type stimulants (ATS), including amphetamine and methamphetamine, rank as the second most commonly abused illicit substances worldwide, trailing only cannabis [5]. In Iraq, Al-Hemiary et al. (2014) confirmed a dramatic surge in illicit ATS use, particularly with methamphetamine and captagon, with a notable increase observed in the Al-Basrah governorate [6,7]. Ciccarone and Shoptaw (2022) discussed a concerning global transition from the third wave of opioid-related overdose deaths to a more critical fourth wave marked by the abuse of methamphetamine and cocaine [8]. Furthermore, Hong et al. (2021) found that synthetic methamphetamine has outpaced traditional drugs like morphine, heroin, and ketamine, becoming one of the most widely abused substances [9]. This shifting landscape underscores the urgent need for comprehensive strategies to address the growing challenges posed by ATS abuse.

Methamphetamine exists as a colorless crystalline solid substance, frequently combined with other chemicals by illicit chemists, while crystal methamphetamine (methamphetamine hydrochloride), the purified form, is the variant commonly smoked or snorted by substance users and addicts [3]. Smoking crystal meth vaporizes the crystals, producing a rapid and intense “high” for users, although this method also renders most of the active form of methamphetamine inert, unlike the oral ingestion route [3,10]. The half-life of smoked crystal meth is approximately eleven hours [2]. In contrast to their illicit use, amphetamines have legitimate medical applications, with physicians utilizing them to manage specific clinical conditions such as narcolepsy and attention deficit hyperactivity disorder (ADHD) [11]. Additionally, amphetamines can serve as adjuvants to antidepressant medications, aiding in managing refractory depression and post-traumatic stress disorder (PTSD) [12,13].

Similarly, experimental trials have explored the potential of certain psychedelics in managing PTSD and treatment-resistant depression [14]. Moreover, emerging evidence supports the use of ATS in managing symptoms associated with traumatic brain injury, stroke, and HIV-related neuropsychiatric manifestations [3]. The medical applications of amphetamines and related substances underscore the importance of further research to optimize their benefits while minimizing potential risks and adverse effects.

Researchers have extensively delved into the intricate web of risk factors that substantially contribute to the development of substance use disorders, with a particular focus on the pernicious grip of crystal methamphetamine addiction. The realm of etiological theories encompasses a multifaceted landscape, spanning biological nuances encompassing genetic predispositions and intricate familial influences. These genetic underpinnings intertwine with the diverse tapestry of addiction-focused models, shedding light on how behavioral patterns can lead down the treacherous path of substance abuse. Intriguingly, the complex contours of personality disorders lie beneath the surface, adding another layer of susceptibility to the lure of substances. Chronic dysregulation, often intertwined with borderline personality disorder, amplifies the propensity for increased impulsivity and an insatiable craving for novelty. When intertwined with the allure of substances, these psychological attributes create a synergy that can escalate into abuse. Beyond the individual scope, a broader canvas emerges when considering psychopathological co-morbidities and environmental influencers. These co-morbidities act as a catalyst, propelling individuals further towards substance misuse. At the same time, the environment in which an individual is nurtured plays a pivotal role. Early substance exposure, catalyzed by triggering life events, psychological stressors, and traumatic experiences, can be a formidable trigger. The resonance of these factors is incredibly potent when combined with the detrimental impact of early social deprivation, which casts a long shadow over an individual’s susceptibility to substance abuse. [15,16,17]. 

The Diagnostic and Statistical Manual of Mental Disorders (DSM-5) outlines specific criteria for diagnosing stimulant use disorder, encompassing crystal meth addiction. Within twelve months, at least two of the following elements must be present: (a) the quantity and duration of stimulant misuse; (b) persistent desire or unsuccessful attempts to control or abstain from stimulant use; (c) spending significant time obtaining, using, and recovering from the effects of the stimulant; (d) experiencing cravings for the stimulant; (e) failing to meet essential role obligations at work, school, or home due to recurrent stimulant use; (f) continued substance use despite persistent or recurrent social and interpersonal problems; (g) giving up on social, occupational, or recreational activities due to stimulant use; (h) engaging in stimulant use in hazardous situations; (i) continuing stimulant use despite knowledge of the associated physical and psychological risks; (j) developing tolerance to the stimulant, requiring increased amounts to achieve the desired effect [18]. These criteria serve as essential diagnostic guidelines to effectively identify and address stimulant use disorder.

The study tackles a pressing societal issue and a prevailing healthcare challenge that has profoundly affected the well-being of countless Iraqis, encompassing individuals struggling with crystal meth abuse and those impacted by suicide and their families during the last two decades. Shedding light on these dimensions, the research offers valuable insights into comprehending the intricate interplay of factors that influence crystal methamphetamine use and its related psychiatric and withdrawal-related manifestations. Ultimately, the study seeks to provide a basis for targeted interventions and support mechanisms designed to assist affected individuals and communities.

### 1.1. Study Aims

Our foremost objective is to discern the risk factors of suicidal ideation among crystal methamphetamine users. This pursuit, however, necessitates a systematic hierarchical approach. Suicidal ideation, as substantiated by existing literature, exhibits intricate links with both psychotic and neurotic attributes in patients. Thus, exploring these factors precedes our examination of the primary objective. Delving into risk factors requires an initial focus on understanding psychotic features and withdrawal manifestations. This sequential progression culminates in our principal goal of comprehending suicidal ideation. Consequently, our secondary objective is identifying risk factors for psychotic features and withdrawal manifestations. This research holds particular significance as it represents the first systematic attempt to investigate and infer potential factors that may predict suicidal tendencies among Iraqi crystal meth users. 

### 1.2. Pre-Study Hypotheses

In our research, we have proposed hypotheses to explain the hierarchical causality model depicting the chain of events leading to suicidal ideation among Iraqi crystal methamphetamine users. Hypothesis 1: Socio-demographic factors, substance abuse characteristics, and triggers may influence or predict the development of psychosis among crystal meth users. Hypothesis 2: Socio-demographic factors and patient characteristics, along with psychotic features, may influence the success or failure of abstinence attempts from crystal meth. Hypothesis 3: Socio-demographic factors, substance abuse characteristics, triggers, psychotic features, and abstinence attempts may collectively impact the withdrawal manifestations experienced by individuals who abstain from crystal meth. Hypothesis 4: The factors mentioned earlier (socio-demographics, substance abuse characteristics, triggers, psychotic features, and abstinence attempts) and the withdrawal features may influence or predict the occurrence of suicidal ideation.

We proposed a multi-domain hierarchy of causality (explained in detail in the methods). In this context, the term “domain” is not synonymous with the term used about questionnaires, along with their associated validity and reliability. 

The collected study variables (socio-demographics, substance abuse characteristics, and triggers) form the first level (Domain 1). The causal chain then progresses up to the psychotic features (Domain 2), abstinence attempts (Domain 3), withdrawal manifestations (Domain 4), and eventually culminates in suicidal ideation at the top level (Domain 5). However, it is essential to acknowledge that the causality hierarchy may not be strictly unidirectional. For instance, suicidal ideations could modulate (either lessen or potentiate) psychotic features and substance abstinence attempts. Complex structural modeling considering the multi-directionality of interactions and additional variables can be more precise in predicting suicidal ideation as the outcome variable [19]. By thoroughly investigating these hypotheses, our research endeavors to unravel crucial insights into the underlying mechanisms contributing to suicidal ideation among crystal meth users from Iraq. These findings could inform targeted interventions and support strategies to address this pressing healthcare issue.

## 2. Materials and Methods

### 2.1. Research Ethics

The current study has received ethical approval from the Department of Psychiatry Ethics Committee, College of Medicine, University of Baghdad. The approval was granted according to protocol number 16 on the 27th of December 2018. Additionally, this study is a part of the research output of a doctorate thesis titled “The use of artificial intelligence or frequentist statistics to navigate the phenomenon of psychedelics (ab)use”. The doctorate thesis was approved by the Bioethics Committee meeting held at the Department of Computer Science and Statistics, Poznan University of Medical Sciences (Uniwersytet Medyczny im. Karola Marcinkowskiego w Poznaniu) in November 2021.

### 2.2. Study Design and Data Collection

The current study utilized a cross-sectional design and employed convenience sampling to recruit Iraqi patients actively abusing crystal methamphetamine and seeking medical assistance at specialized addiction clinics. The participants were recruited from Baghdad Teaching Hospital and Ibn-Rushd Teaching Hospital. Data collection took place between February and October 2021. 

The survey was crafted by a panel of experts in psychiatry, including a consultant and two senior specialist psychiatrists. Their collective expertise in addiction psychiatry guided the survey development. They astutely selected a comprehensive range of variables, responding to the criteria stipulated in DSM-5 for stimulant use disorder (SUD). This strategic approach not only lends depth and a cogent basis for elucidating the factors underlying suicidal ideation among Iraqi users of crystal methamphetamine. While acknowledging that the preceding explanation draws on anecdotal perspectives possessing a lower level of evidence, it remains firmly rooted by addressing well-established diagnostic criteria for SUD. 

On the other hand, it is worth noting that the survey administration took a structured approach, employing interviews conducted by two adept specialist psychiatrists (raters) well-versed in the intricacies of managing individuals grappling with crystal methamphetamine addiction. This method of administration ensured a reliable and unbiased assessment, eschewing potential self-reporting biases. The survey meticulously charted many variables, overseen by the same diligent psychiatrists who validated the data. When differences in interpretation arose between the two raters, the consultant psychiatrist interfered as a third rater to adjudicate and resolve the discrepancies.

The survey mapped and gathered data on the participant’s substance abuse patterns, symptoms, and other relevant factors that align with the diagnostic criteria.

### 2.3. Study Participants

The minimum sample size was calculated for the population proportion based on the finite population formula while considering the prevalence of crystal meth use in Iraq (0.6%), the significance level (α = 0.05), and the margin of error (5%). Based on this calculation, a sample size of 384 patients was determined. However, the study could only recruit 165 patients due to limited availability, leading to a smaller sample size than initially intended. Although a post hoc power analysis was conducted to validate the statistical robustness, the smaller sample size remains a study limitation [20].

Further, the current research employed a specific exclusion criterion, where patients with dual diagnoses, such as those with both substance use disorders and other neuropsychiatric disorders, were excluded from the study. A collective of 39 patients with dual diagnoses were excluded.

### 2.4. Study Variables

Using the survey, data collection encompassed several key variables representing different aspects of the study objectives: (1) Sociodemographics: This category included information about participants’ age, gender, educational background, marital status, employment status, and other relevant characteristics. (2) Substance abuse severity and chronicity: The researchers collected data on the severity of crystal methamphetamine use among the participants, represented by the duration and chronicity of their substance abuse. (3) Clinical manifestations: This category assessed participants for various clinical features, including psychotic and neurotic manifestations. Psychotic features include hallucinations and delusions, while neurotic features involve anxiety, depression, or other neurotic symptoms. (4) Substance abuse trigger events: The researchers inquired about specific events or circumstances that might have triggered or influenced the initiation or escalation of crystal methamphetamine use among the participants. 

By analyzing these variables, the study aimed to gain comprehensive insights into the risk factors and predictors associated with psychotic features, withdrawal manifestations, and suicidal ideation among individuals solely diagnosed with crystal methamphetamine use disorder in Baghdad, Iraq. The assignment of variables into independent (explanatory variables) and dependent (outcome variables) categories is not arbitrary; it directly relates to the “Pre-study hypothesis” section we discussed earlier. Moreover, we have considered the renowned Bradford Hill’s criteria on causality to guide our variables’ assignment. 

The socio-demographic parameters undoubtedly represent the most fundamental variables; some are inherent and genetically determined (e.g., ethnicity, gender, among others) or controlled by time (e.g., age, disease progression, among others). These parameters represent the foundational elements (Domain 1) that often influence other variables. We have already highlighted that the direction of causality interaction could be bidirectional or reciprocal rather than unidirectional. Moving upward in the hierarchy, we recognize that suicidal ideation (or suicide) represents the ultimate outcome (Domain 5), with the other variables occupying intermediary positions. Specifically, it is noteworthy that psychotic features (Domain 2) typically precede the first abstinence attempts (Domain 3) and withdrawal manifestations (Domain 4), establishing an undeniable chronological (temporal) and causal sequence. 

### 2.5. Criteria for Suicidal Ideation

In our study, we utilized Beck’s Suicide Ideation Scale-Current (SSI-C) questionnaire to assess and score suicidal ideation among the participants. The questionnaire assigns scores ranging from 0 to 38, with higher scores indicating a stronger inclination towards higher-risk suicidal ideation (suicidal ideators) as opposed to lower-risk suicidal ideation (non-ideators) [21,22]. We adopted the threshold established in the original Beck study to determine the cut-off point for distinguishing between these two groups (Beck et al., 1999). According to this study, a raw total score of 2 or higher on Beck’s SSI-C signifies a higher risk of suicidal ideation (suicidal ideators) than a raw total score of less than 2, which indicates a lower risk (non-ideators). The cut-off score’s sensitivity rate was 53%, while the specificity rate was 83% [23]. By implementing this cut-off point, we aimed to categorize participants into distinct groups based on their level of suicidal ideation, thus facilitating the analysis of risk factors and predictors associated with suicidal tendencies among crystal methamphetamine users in our study.

### 2.6. Data Analysis

For data analysis, the researchers tabulated the collected data using Microsoft Excel 2016 and imported it into TIBCO Statistica version 13.3 (2018) for further analysis. A receiver operating characteristic curve (ROC) analysis was employed to investigate the effect of age on psychotic features, abstinence attempts, and suicidal ideation. The results were presented as the area under the curve, and the optimal cut-off point was identified using the Youden index. This optimal cut-off point was provided with statistical accuracy and predictive values (negative and positive).

Multivariable logistic regression was utilized to identify risk factors for psychotic features, abstinence attempts, and suicidal ideation. The results were presented as odds ratios (OR) based on the exponentiated B coefficients. Binary logistic regression was chosen for its superior accuracy [24]. The study conducted three multivariable logistic regression models based on the pre-selection of potential risk factors. The first model incorporated all significant risk factors identified during univariable analysis. The second model included all potential risk factors and underwent a backward stepwise selection. The third model relied on the pre-selection of significant risk factors from model 1 and model 2. It demonstrated the highest statistical accuracy among all models and was considered optimal. The goodness-of-fit of the logistic regression models was assessed using the Hosmer and Lemeshow test (HL test). Nagelkerke’s R^2^ was reported as a measure of the proportion of the total variation in the dependent variable explained by the independent variables. 

## 3. Results

### 3.1. Patients’ Attributes, Socio-Demographics, History of Illnesses, and Other Parameters

The total sample included 165 patients (n = 165), including males (n_m_ = 152, 92.1%) and females (n_f_ = 13, 7.9%). Most individuals were in the third decade of life (26.62 ± 0.53). The patients exhibiting suicidal ideations displayed a slightly higher mean age than those without (27.42 ± 1.11 versus 26.26 ± 0.58) (Table 1). The patient population was divided regarding marital status, with 48.5% married and 51.5% single. The majority of patients had attained a primary or intermediate level of education (79.4%), while those in a smaller proportion were either illiterate (5.5%) or had achieved a secondary school or college-level education (15.2%). Furthermore, a significant majority of patients resided in urban areas (94.5%), with a considerable portion employed in blue-collar occupations (81.2%) or serving in military, militant, and security forces (10.3%); the remaining patients comprised unemployed individuals or students (8.5%). Regarding crystal meth usage, a substantial proportion (61.8%) reported using the substance for over one year. Additionally, most patients (67.9%) reported daily usage, with the remaining individuals using it weekly or monthly. The preferred administration methods varied, with 55.8% choosing to snort crystal meth, 43% engaging in both oral and snorting administration, and a minor percentage (1.2%) employing both oral and smoking methods. Notably, patients reported using crystal meth either alone (55.2%), in combination with alcohol (24.8%), or in conjunction with other drugs (20%), often in social settings with peers (30.3%) or in solitude (69.7%).

Regarding the monthly expenditure on purchasing crystal meth, approximately two-thirds of individuals allocated around USD 200 (64.2%) or USD 200–400 (14.5%) for this purpose. A smaller proportion either acquired the substance without monetary expenditure, receiving it from friends and family (7.9%), or spent approximately USD 400–600 per month (7.3%), while only a minority (6.1%) exceeded monthly spending of USD 600. The acquisition of crystalline methamphetamine was primarily accomplished through interactions with drug dealers (53.9%) or associations with friends (46.1%). Approximately two-thirds of users (69.7%) attempted to abstain from using crystal meth, while around one-third (35.8%) reported no withdrawal signs and symptoms following abstinence. In contrast, the remaining individuals experienced either somatic withdrawal manifestations (32.1%), such as tremors, generalized body aches, and fatigue, or neurotic manifestations (29.1%). A smaller fraction (3%) suffered from psychosis during withdrawal.

Concerning neurotic manifestations, a considerable majority (73.3%) of users reported episodes of aggression, while the vast majority experienced sleep disturbances (92.7%), anorexia (90.3%), and weight loss (80%). Additionally, a significant proportion of users (63.6%) reported associated mood disturbances, predominantly neurotic, with a smaller fraction experiencing social problems or co-disturbances, such as mood swings and social disruptions. Conversely, a substantially smaller group (36.4%) reported associated physical illnesses, primarily as joint aches, generalized weakness, and palpitations. A minority of individuals experienced tooth loss and viral hepatitis linked to crystal meth use. 

Regarding psychotic manifestations, a minority of patients (40.6%) reported auditory hallucinations, while a similar proportion (41.8%) experienced visual hallucinations. However, more patients (54.5%) exhibited persecutory delusions, and a substantial majority (72.1%) suffered from delusions of infidelity. Regarding potential stimulants (triggers) for the initiation of crystal meth use, stressful events (64.2%), psychological trauma (61.8%), and peer pressure (56.4%) were identified as triggering factors linked to substance abuse initiation. Family problems, financial hardship, and marital conflicts were also prevalent, affecting 62.4%, 86.1%, and 66.1% of the sample. Furthermore, almost two-thirds of users (66.1%) were driven by curiosity about crystal meth. Additionally, approximately three-quarters of patients (73.9%) reported the presence of another illicit substance user within the same family. On the contrary, a positive family psychiatric history was observed in only a smaller subset of patients (15.2%).

### 3.2. Inference concerning Age

Most individuals fell within the mid-third decade of life, with a mean age of 26.62 ± 0.53 years, while those with suicidal ideations were slightly older than others (27.42 ± 1.11 versus 26.26 ± 0.58). However, despite the observed difference in mean age, statistical analysis revealed no significant difference between the two groups. Furthermore, age did not show significant variations based on gender or the frequency of substance use. Regarding the presence of psychotic features, we compared age distribution across categories of visual hallucinations, auditory hallucinations, persecutory delusions, and delusions of infidelity; however, no significant differences were found. Similarly, when we examined age distribution across categories of substance abstinence attempts, the Kruskal–Wallis H test yielded non-significant results (H = 1.27, *p* = 0.530). Additionally, no significant differences in age were observed across categories of withdrawal manifestations and suicidal ideations.

Regarding the ROC analysis, we thoroughly interpreted the area under the curve (AUC) and its significance while examining the influence of age on psychotic manifestations, substance abstinence attempts, withdrawal manifestations, and suicidal ideation. Interestingly, we identified a significant age effect specifically within female substance users, impacting four specific outcome variables: auditory hallucinations (AUC = 0.94 ± 0.07, *p* < 0.001, statistical accuracy = 82%), abstinence attempts (AUC = 0.85 ± 0.11, *p* = 0.002, 63%), withdrawal manifestations (AUC = 0.85 ± 0.11, *p* = 0.002, 63%), and suicidal ideation (AUC = 0.97 ± 0.042, *p* < 0.001, 89%). Regarding the age cut-off criterion, it led to the segmentation of female patients into two distinct age cohorts: those below 20 years of age and those aged 20 years or above. Although the ROC analyses exhibited statistical significance and commendable accuracy, we computed each outcome variable’s positive and negative predictive values (PPVs and NPVs) for further verification. 

It is imperative to acknowledge that the relatively restricted representation of females within our study (constituting 7.9% of the total sample) raises concerns about the robustness of the statistical inferences, particularly in the context of broader population samples. Turning our attention to predictive values, their efficacy varied. In the realm of prognosticating auditory hallucinations, their utility is useless (PPV = 39.01%; NPV = 41.67%). Conversely, these values prove somewhat beneficial in identifying patients devoid of suicidal ideations (PPV = 32.62%; NPV = 75.00%). Furthermore, they demonstrate limited practicality in discerning individuals who opt for abstention from crystal methamphetamine (PPV = 68.79%; NPV = 25.00%), along with those susceptible to developing withdrawal manifestations following abstinence (PPV = 63.83%; NPV = 33.33%). The former results indicate low practicality in predicting the studied outcomes based on age among female patients.

### 3.3. Logistic Regression Analysis

#### 3.3.1. Domain 5: Suicidal Ideations 

Regarding suicidal ideation, the model successfully passed the Hosmer and Lemeshow (HL) test for the goodness-of-fit assessment (*p* = 0.885). The model demonstrated a favorable statistical accuracy (79.4%). Among the identified risk factors (Table 2), seven variables were identified, listed here in descending order of strength: use duration (OR = 6.15, *p* = 0.001), absence of psychological trauma (OR = 4.58, *p* = 0.006), visual hallucinations (OR = 4.52, *p* = 0.001), aggression episodes (OR = 3.88, *p* = 0.016), family member users (OR = 3.61, *p* = 0.016), use environment (OR = 3.35, *p* = 0.022), and the concomitant use of other substances (OR = 3.23, *p* = 0.033). In summary, individuals who use crystal meth and exhibit suicidal ideation are more likely to have the following characteristics: they are chronic users (more than one year), they have not experienced psychological trauma, suffer from visual hallucinations, experience aggression episodes, have a family member who uses illicit substances, use crystal meth in social settings with peers, and they engage in the abuse of other illicit substances (poly-drug abuse) in addition to alcohol.

#### 3.3.2. Domain 4: Withdrawal Manifestations

The model demonstrated a satisfactory result in the HL test (*p* = 0.980) and good statistical accuracy (70.9%). Three significant risk factors were found: poly-drug abuse, specifically with alcohol (OR = 5.64, *p* = 0.004), the presence of associated mood disturbances (OR = 2.88, *p* = 0.004), and prolonged use duration of crystal methamphetamine (OR = 2.31, *p* = 0.023) (Table 2). In summary, individuals who use crystal meth and experience withdrawal manifestations are more likely to do so if they engage in poly-drug abuse, particularly with alcohol, exhibit associated mood disturbances, and have a chronic history of substance use lasting more than one year.

#### 3.3.3. Domain 3: Abstinence Attempts

The model exhibited a favorable goodness-of-fit (*p* = 0.417) and a good statistical accuracy (75.8%). Significant risk factors included: associated physical illnesses (OR = 5.80, *p* = 0.001), poly-drug abuse involving alcohol (OR = 3.78, *p* = 0.055) and other illicit drugs (OR = 2.87, *p* = 0.031), the occurrence of auditory hallucinations (OR = 2.54, *p* = 0.027), and prolonged use of crystal meth (OR = 2.33, *p* = 0.036) (Table 2). In summary, individuals who use crystal meth are more likely to attempt abstinence from the substance when they suffer from associated physical illnesses, engage in multiple substance abuse, experience auditory hallucinations, and have chronically used crystal meth for more than one year.

#### 3.3.4. Domain 2a: Visual Hallucinations

The model satisfied the HL test (*p* = 0.726) and demonstrated high statistical accuracy (87.3%). Five risk factors existed: persecutory delusions (OR = 4.34, *p* = 0.009), exposure to stressful events (OR = 3.69, *p* = 0.017), presence of suicidal ideation (OR = 3.42, *p* = 0.018), exploratory drive (OR = 3.50, *p* = 0.029), and frequency of substance use (OR = 2.73, *p* = 0.065) (Table 3). Individuals who use crystal meth are more likely to experience visual hallucinations when they have persecutory delusions, encounter stressful events, have suicidal ideations, possess curiosity towards crystal meth, and engage in daily crystal meth use.

#### 3.3.5. Domain 2a: Auditory Hallucinations

The regression model for auditory hallucinations yielded comparable results to the visual hallucinations analysis regarding robust statistical accuracy (85.5%), and it also successfully satisfied the HL test (*p* = 0.214). However, it revealed distinct and fewer risk factors than those associated with visual hallucinations (Table 3). These factors included male gender (OR = 4.57, *p* = 0.069), use duration (OR = 2.59, *p* = 0.055), and exploratory drive (OR = 2.26, *p* = 0.090). In summary, individuals who use crystalline methamphetamine are more likely to experience auditory hallucinations if they are males, have a chronic history of methamphetamine (lasting more than one year), and possess curiosity towards crystal meth. 

#### 3.3.6. Domain 2b: Persecutory Delusions

The model for persecutory delusions also demonstrated a satisfying goodness-of-fit (*p* = 0.627) and exhibited reliable statistical accuracy (83%). The model identified five significant risk factors: occupation (military and security forces: OR = 7.33, *p* = 0.016; students and unemployed individuals: OR = 4.24, *p* = 0.066), delusions of infidelity (OR = 5.23, *p* = 0.003), visual hallucinations (OR = 5.19, *p* < 0.001), educational level (primary or intermediate school: OR = 4.70, *p* = 0.030), and associated mood disturbances (OR = 3.34, *p* = 0.009) (Table 3). In summary, individuals who abuse crystal meth are more prone to experiencing persecutory delusions if they fall under specific occupation categories, such as students, unemployed individuals, or military and security forces employees. Additionally, they may suffer from coexisting delusions of infidelity concerning their spouses or significant others, experience visual hallucinations, possess a basic education, and experience other associated mood disturbances.

#### 3.3.7. Domain 2b: Delusions of Infidelity

The binary logistic regression analysis for delusions of infidelity demonstrated good statistical accuracy (83%) and satisfactory results in the goodness-of-fit test (*p* = 0.059). However, compared to the persecutory delusion model, it exhibited a lower sensitivity (65.2% vs. 85.6%) but a higher specificity (89.9% vs. 80%). The model revealed four distinct risk factors: experiencing marital quarrels (OR = 7.70, *p* < 0.001), engaging in poly-drug abuse with alcohol (OR = 6.37, *p* = 0.002) and other illicit substances (OR = 2.88, *p* = 0.043), having persecutory delusions (OR = 6.20, *p* < 0.001), and having another family member who uses illicit substances (OR = 3.70, *p* = 0.026) (Table 3). In summary, individuals who use crystal methamphetamine are more susceptible to experiencing delusions of infidelity when they frequently encounter marital quarrels, engage in poly-drug use, specifically involving alcohol and other illicit substances, experience coexisting persecutory delusions, and have another family member who uses illicit substances. 

### 3.4. Integrative Interpretation of the Pre-Study Hypotheses and Multivariable Analysis

The current study proposed a hierarchical causal relationship among the different domains of variables, with each level influencing the higher domains (Figure 1). This progression started from Domain 1 (socio-demographics, substance abuse parameters, and triggers of use), moving through Domain 2 (psychotic features), then to Domain 3 (substance abstinence attempts), Domain 4 (withdrawal manifestations), and ultimately reaching Domain 5 (suicidal ideation). The findings indicate some congruence between the pre-study hypotheses and the multivariable analyses. A few reciprocal relationships existed, such as between visual hallucinations and suicidal ideations. Further, some risk factors from one domain unexpectedly transcended over to influence higher domains, such as risk factors from the first domain influencing substance abstinence, withdrawal manifestations, and suicidal ideation. On the other hand, the study could not establish a direct causal relationship between Domains 3 and 4 (substance abstinence and withdrawal manifestations) and Domain 5 (suicidal ideation). 

The study also observed that as it progressed from the lowest domain to the highest, the statistical accuracy of the models declined, particularly beyond the domain of psychotic features (Table 4). This finding suggests that the models became less robust in predicting the highest-domain outcome (suicidal ideation) and the lower-domain variables (abstinence attempts and withdrawal manifestations). 

## 4. Discussion

The study highlights the need for further research using larger sample sizes to enhance the reliability of the findings. Re-evaluating risk factors disregarded in earlier studies and exploring hidden variables should be considered. Additionally, the study suggests including genetic and ethnic factors to understand better the complex interplay between the risk factors in crystal methamphetamine users. Crystal meth users are at a higher risk of experiencing suicidal ideations if they meet the following conditions: they are chronic users of crystal meth for more than one year, have no psychological trauma, suffer from visual hallucinations, experience aggression episodes, have family members who use illicit substances, use crystal meth in social settings with peers, and abuse other illicit substances alongside crystal meth, excluding alcohol.

The correlation between psychological trauma and suicidal ideation presents a paradoxical association. Notably, an intriguing phenomenon emerges wherein the absence of psychological trauma contributes to suicidal ideation among individuals using crystal methamphetamine in Iraq. This perplexing connection, however, can be elucidated through careful examination. Firstly, it is crucial to recognize that psychological trauma frequently manifests as an acute event, while subsequent psychopathological outcomes such as post-traumatic stress disorder tend to evolve into chronic processes. This chronicity can lead to a desensitization effect, rendering individuals less responsive to subsequent traumatic occurrences. Secondly, the manifestation of PTSD often involves an avoidance response towards situations that could potentially evoke trauma or pose life-threatening circumstances, mirroring elements of the original traumatic experience. Thirdly, this adaptive mechanism could conceivably bolster the self-preservation instincts in individuals afflicted by PTSD, consequently reducing the susceptibility to suicidal ideation despite their history of psychological trauma. In contrast, individuals devoid of antecedent psychological trauma who abuse crystal methamphetamine might exhibit heightened vulnerability to such suicidal ideations [25,26]. This intricate interplay demands further investigation to investigate the elaborate dynamics at play comprehensively.

On the other hand, Iraqi crystal meth users are more likely to experience withdrawal manifestations if they are poly-drug abusers of alcohol and other illicit substances, exhibit associated mood disturbances, and have chronically abused crystal meth for over one year. On the contrary, patients may have a higher chance of successfully abstaining from crystal meth use if they suffer from associated physical illnesses, engage in poly-drug abuse (including alcohol), experience auditory hallucinations, and have been long-lasting users of crystal meth for more than one year.

Regarding psychotic features among crystal meth users, they are more likely to experience visual hallucinations under the following conditions: when they suffer from persecutory delusions, encounter stressful events, have suicidal ideations, exhibit an exploratory drive (curiosity) towards crystal meth, and use the substance daily. Conversely, crystal meth users are more prone to auditory hallucinations if they are males, chronic users (using for more than one year), and demonstrate an exploratory drive towards using the substance. Regarding delusions, methamphetamine abusers are more susceptible to experiencing persecutory delusions if they fall into the following categories: they are military or security force employees, unemployed or students, suffer from coexisting delusions of infidelity concerning their spouses or significant others, experience visual hallucinations, have a primary or intermediate school education, or are illiterate. 

Our study did not determine the specific causes of persecutory delusions among students and unemployed methamphetamine users. The psychopathology of delusional subtypes is very complex. It would be necessary to conduct studies focused exclusively on one group to clarify all the factors that can generate delusions of this type and others. However, when it comes to military and security forces personnel who, for various reasons, took amphetamines in the course of their professional duties, one can find information in scientific publications that may suggest predictors of delusions in this group [27,28]. However, this needs to be confirmed in research, which in this hermetically sealed group seems arduous.

Furthermore, Iraqi crystal meth users are more likely to experience delusions of infidelity under the following conditions: experiencing marital quarrels, being poly-drug users (using alcohol and other illicit substances), suffering from coexisting persecutory delusions, and having family members who use illicit substances and alcohol.

Based on our findings, our logistic models demonstrated higher precision in predicting persecutory delusions than delusions of infidelity. On the other hand, the top risk factors associated with delusions of infidelity (marital quarrels, poly-drug abuse, and persecutory delusions) are distinct from those linked to persecutory delusions. Moreover, the nature of users’ occupations, especially military forces employees, seems to render them more susceptible to experiencing persecutory delusions rather than delusions of infidelity. This observation may warrant further investigation to better understand the underlying factors contributing to these differences in delusional experiences among crystal methamphetamine users.

It is essential to consider the socio-political environment of the region where our study was conducted while attempting to identify risk factors and predictors of psychotic symptoms, substance abstinence attempts, withdrawal symptoms, and suicidal thoughts or attempts among Iraqi methamphetamine users. In Iraq, dynamic social processes such as conflicts and wars have persisted for several decades, leading to significant stress among the population [6]. Factors such as high unemployment rates, awareness of corruption among decision-makers, protests in “Al-Tahrir” (the Liberation) Square in Baghdad against living conditions, violence against unarmed demonstrators, and the widespread experience of poverty contribute to the challenging circumstances faced by many Iraqis [29]. These socio-political factors can significantly impact mental health outcomes and substance abuse patterns among crystal methamphetamine users in Iraq. The study’s findings should be considered within these broader social dynamics to comprehend better the complexities and challenges faced by individuals struggling with addiction and mental health issues in this region.

As early as 2006, the study authors noted that following the United States invasion of Iraq, there was a notable increase in indications of drug use, particularly among soldiers, police officers, and other security forces. The unpredictable reality and the trauma (PTSD) resulting from frequent encounters with death in the environment, combined with the prohibition of alcohol consumption, drove individuals to seek relief through alternative means [30]. Additionally, the post-invasion period saw a rise in corruption, economic instability, increased social inequality, and cuts in public spending, leading to challenging daily living conditions for the population [31]. The difficult economic situation profoundly impacted Iraqis’ emotional and mental states, further compounded by the precarious state of healthcare. In 2008, a prominent scholarly journal reported on the alarming situation of Iraq’s healthcare system, which included damaged and looted healthcare facilities, the voluntary departure of healthcare workers, especially physicians, from the country, persecution, kidnappings, and the killing of around 2000 doctors during hostilities [32]. These conditions of constant tension were intensified by uncertainty, economic hardships, and diminishing access to proper healthcare, including mental health services.

The high demand to alleviate distressing emotional states is evident through illegal attempts to obtain mood-altering pharmaceuticals, as seen in widespread pharmacy looting and fraud in pharmacies [33,34]. Additionally, it is vital to highlight the limited capacity of Iraqi psychiatry facilities and resources, including insufficient hospitals and mental health clinics, a shortage of psychiatrists, limited medication availability, lack of widespread knowledge about psychiatric help options, and reliance on non-standard treatments such as “faith healing” for mental disorders [29]. These factors act as predictors that make attaining emotional stability challenging. While substance abuse has historically been considered relatively low in Iraq, poverty and uncontrolled borders have increased indications of drug use, particularly in the past decade. The use of crystal methamphetamine has garnered exceptional interest among potential users and concern among researchers studying the phenomenon [29]. 

The scarcity of studies on Iraqi methamphetamine users, particularly concerning crystal meth, highlights the significant value of our current research. One survey conducted in 2019 with 600 Iraqi drug users aged 15–60 revealed that methamphetamine is the most popular drug, with 32% of respondents admitting to its use. Although this study looked at various drugs, it is challenging to pinpoint specific demographic characteristics that may predict methamphetamine use based on the data. However, it can be inferred from the data that drug use is more prevalent among adults aged 19 and older, slightly more likely among those who are married, have children, possess a primary or middle (intermediate) school education, have employment, and live in their households [35], which suggests that adults who feel responsible for their loved ones’ well-being may be more prone to using drugs, possibly due to the anxiety associated with meeting these demands. Although these are speculations made by the study authors, it is worth noting and validating these conclusions through prospective investigations. Further research is needed to comprehensively understand methamphetamine use patterns and their underlying factors among Iraqi users, particularly in the context of crystal methamphetamine.

The results obtained in our study align with data published in world opinion journals. According to these publications, drugs, particularly methamphetamine, are predominantly used by young and unemployed individuals [36]. Data from Al-Basrah police department further support this trend, showing that 97% of drug users arrested in 2018 were unemployed, with over two-thirds of them being young people under 25 years of age [37].

The combination of unemployment, corruption affecting the traditional way of living in Iraq, the proliferation of criminal organizations and militias, and the weak judicial system is considered predictive of the increasing drug consumption phenomenon, including crystal methamphetamine, in the country [38]. Additionally, individuals who have served time in prison due to drug-related crimes often face difficulties in finding legal and decent employment opportunities with fair pay. In Iraq, a drug conviction is considered a “dishonorable offense,” leading employers to be reluctant to hire ex-convicts due to the stigma attached to them [36]. This situation may contribute to a relapse of drug use to cope with unpleasant emotional states while also exacerbating the health consequences associated with methamphetamine use. These findings highlight the complex interplay of socio-economic factors and their impact on drug use patterns in Iraq, emphasizing the need for comprehensive approaches to address the root causes of substance abuse in the country.

In light of the aims of our study, which focused on identifying socio-demographic characteristics, clinical factors associated with methamphetamine use, and factors contributing to the initiation of crystal meth use, it is essential to acknowledge that the available literature highlights several recurring circumstances and risk factors: Drug use, including methamphetamine, is prevalent among adolescents and young adults, typically ranging from 15 to 35 years of age, with addiction centers commonly seeing individuals aged between 17 and 25 years. Some reports indicate that more than 40% of people in certain age groups in Iraq use drugs [39].Methamphetamine use is also reported among young women who use the drug to cope with the daily pressures of life and the emotional challenges of domestic violence [39].Methamphetamine use has been associated with experiencing paranoid states, engaging in violent acts, and committing crimes. It is also linked to various behavioral disorders within the anxiety spectrum, other neurotic conditions, and aggression [40,41].The city of Al-Basrah, with its four million inhabitants, is described as “flooded” with methamphetamine, but the available treatment facilities for addicts are limited, with only 44 hospital beds available. The treatment facilities often lack proper organization and resemble isolation in a prison-like setting. The prevailing political crises, unemployment, and corruption create an environment where methamphetamine becomes an escape from reality for many vulnerable young individuals [39,42,43].The proliferation of “drug cafés” in Iraq, where water pipes are spiked with drugs, possibly without the customer’s knowledge, contributes to the drug use phenomenon [39]. The economic impact of the COVID-19 pandemic has also worsened poverty, leading some to seek relief from drugs [40].Numerous online articles emphasize the urgent need for research to determine the precise scale and nationwide prevalence of the drug use phenomenon in Iraq. Identifying predictors of such behavior is crucial to developing adequate preventive strategies, interventional protocols, and harm and risk reduction measures to address the alarming situation in the country.

Given these recurring circumstances and risk factors, our study plays a significant role in understanding methamphetamine use among Iraqis. It provides valuable insights into potential predictors and factors associated with its use in this context. By recognizing these patterns and understanding the underlying factors, we can work towards implementing effective interventions and preventive measures to address the drug use issue in Iraq.

Research on the link between substance use and suicide has gained increasing attention in many developed countries [41,44]. However, this study area is almost ignored in middle- and low-income countries. This gap in research may hinder the effectiveness of suicide prevention efforts in these regions, as strategies based on models used in highly developed countries may not fully consider the unique challenges and factors prevalent in middle- and low-income countries. Therefore, research considering the economic and social context is essential to develop more effective prevention measures [41]. One of the reasons for the lack of research on the link between crystal methamphetamine use and suicide in middle- and low-income countries could be the underreporting of drug use in these regions, as well as legal restrictions and a lack of expertise in screening for drug use [41]. These factors may contribute to the limited availability of accurate data and awareness of the issue, making it challenging to address the problem effectively. In conclusion, promoting research on the association between substance use, particularly crystal methamphetamine, and suicide in middle- and low-income countries is of utmost importance [41,44]. By understanding the specific challenges and factors involved, tailored prevention strategies can be developed to address this pressing public health concern effectively.

### Study Limitations 

The current study has several limitations that need to be acknowledged. Firstly, the sample size was relatively small, which may limit the generalizability of the findings. The researchers calculated the minimum sample size based on the prevalence of crystal meth use and suicidal ideation in Iraq [45], but due to practical constraints, only 165 patients were recruited. Similarly, the relatively small representations of females (9.09%) pose a contributing factor that may limit the generalizability of the ROC analysis findings. 

Furthermore, it is essential to note that the SSI-C scale utilized for evaluating suicidal ideation exhibits some variability in sensitivity and specificity. This variability might have a modest influence on its accuracy in distinguishing between individuals with and without suicidal ideation. Another aspect to consider is the potential impact of varying cut-off scores across different studies, which has been observed to vary in some cases [22,46]. 

Notwithstanding these constraints, the research offers significant insights into the risk factors linked with crystal meth usage and its effects on mental health results. Conversely, our study was rigorous by excluding individuals with dual diagnoses, especially those experiencing substance use disorders and severe mental or neuropsychiatric conditions [47,48]. Although this exclusion proved essential, forthcoming investigations encompassing more extensive samples with diverse strata and subgroups, including individuals with dual diagnoses, could substantially augment our comprehension of this intricate concern.

## 5. Conclusions

The study findings unraveled the presence of risk factors associated with each outcome variable among Iraqi crystal methamphetamine users. The main risk factors include (1) duration of use exceeding one year for suicidal ideation, (2) poly-drug abuse for withdrawal manifestations, (3) associated physical illnesses for substance abstinence attempts, (4) persecutory delusions for visual hallucinations, (5) male gender for auditory hallucinations, (6) unemployment and military personnel status for persecutory delusions, and (7) marital quarrels for delusions of infidelity. The presence of these risk factors highlights the necessity for an interdisciplinary approach involving psychiatrists, clinical psychologists, and social workers to evaluate at-risk individuals abusing crystal methamphetamine. By assessing and monitoring these individuals, professionals can intervene to prevent their progression within the suicide spectrum, which may encompass ideation, parasuicide, and ultimately completed suicide. 

Particular attention should be given to the duration of crystal methamphetamine use, co-occurring poly-drug abuse, individuals’ employment status, and interpersonal relationships within spousal contexts. It is important to emphasize that these findings are specific to Iraqi crystal methamphetamine users and relate to a critical societal issue and highly pervasive healthcare problem that has profoundly affected the quality of life for many Iraqis, including individuals struggling with substance abuse, those experiencing suicidal tendencies, and their families, over the past two decades. The study’s insights provide valuable information for addressing and mitigating the impact of crystal methamphetamine use on mental health and overall well-being in the Iraqi context.

## Figures and Tables

**Figure 1 brainsci-13-01279-f001:**
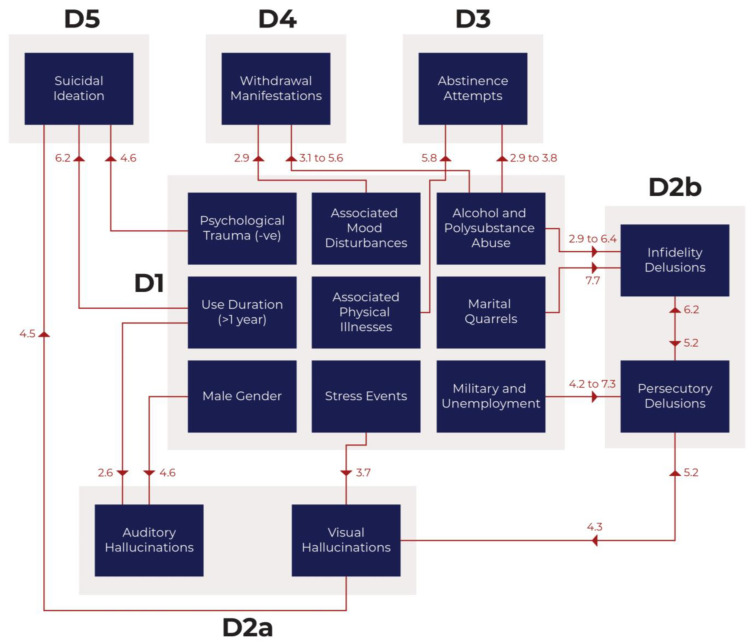
The hierarchical causal model.The numbers represent the odds ratio for the top risk factors.

**Table 1 brainsci-13-01279-t001:** Descriptive statistics of age based on patients’ attributes.

Attribute	Status	N	Range	Minimum	Maximum	Mean ± SEM
Suicidal ideation	No	113	30	15	45	26.26 ± 0.58
	Yes	52	35	17	52	27.42 ± 1.11
Withdrawal manifestations	No	59	32	15	47	25.51 ± 0.77
	Yes	106	35	17	52	27.25 ± 0.70
Abstinence attempts	No	50	25	15	40	25.44 ± 0.74
	Yes	115	36	16	52	27.14 ± 0.69
Auditory hallucinations	No	96	31	16	47	26.35 ± 0.57
	Yes	69	37	15	52	27.00 ± 0.99
Visual hallucinations	No	98	36	16	52	27.02 ± 0.66
	Yes	67	30	15	45	26.04 ± 0.88
Persecutory delusions	No	75	31	16	47	26.88 ± 0.77
	Yes	90	37	15	52	26.41 ± 0.73
Infidelity delusions	No	119	29	16	45	26.12 ± 0.55
	Yes	46	37	15	52	27.93 ± 1.27

SEM: Standard error of the mean.

**Table 2 brainsci-13-01279-t002:** The optimal model’s odds ratio and confidence interval.

Outcome Variable	Risk Factors		*p*-Value	Odds Ratio	95% CI	
					Lower Bound	Upper Bound
Suicidal Ideation (Domain 5)	Use Duration (>1 Year)		0.001	6.15	2.15	17.53
	Psychological Trauma (No)		0.006	4.58	1.54	13.62
	Visual Hallucinations (Yes)		0.001	4.52	1.88	10.90
	Aggression Episodes (Yes)		0.016	3.88	1.28	11.73
	Family Member Users (Yes)		0.016	3.61	1.27	10.28
	Use Environment (with peers)		0.022	3.35	1.19	9.39
	Other Substances	Other Drugs	0.033	3.23	1.10	9.48
		Alcohol	0.455	0.62	0.18	2.15
		Alone				
Withdrawal Manifestations (Domain 4)	Other Substances	Alcohol	0.004	5.64	1.76	18.08
		Other Drugs	0.011	3.09	1.30	7.35
		Alone				
	Associated Mood Disturbances (Yes)		0.004	2.88	1.40	5.93
	Use Duration (>1 Year)		0.023	2.31	1.12	4.78
Abstinence Attempts (Domain 3)	Associated Physical Illnesses (Yes)		0.001	5.80	2.01	16.79
	Other Substances	Alcohol	0.055	3.78	0.97	14.74
		Other Drugs	0.031	2.87	1.10	7.49
		Alone				
	Auditory Hallucinations (Yes)		0.027	2.54	1.11	5.78
	Use Duration (>1 Year)		0.036	2.33	1.06	5.13

When a variable has three categories, the last represents the reference category.

**Table 3 brainsci-13-01279-t003:** The optimal model’s odds ratio and confidence interval.

Outcome Variable	Risk Factors		*p*-Value	Odds Ratio	95% CI	
					Upper Bound	Lower Bound
Visual Hallucinations (Domain 2)	Persecutory Delusions (Yes)		0.009	4.34	1.44	13.06
	Stressful Event (Yes)		0.017	3.69	1.27	10.76
	Suicidal Ideation (Yes)		0.018	3.42	1.24	9.42
	Exploratory Drive (Yes)		0.029	3.50	1.14	10.74
	Use Frequency (Daily)		0.065	2.73	0.94	7.94
Auditory Hallucinations (Domain 2)	Gender (Male)		0.069	4.57	0.89	23.48
	Use Duration (>1 Year)		0.055	2.59	0.98	6.84
	Exploratory Drive (Yes)		0.090	2.26	0.88	5.79
Persecutory Delusions (Domain 2)	Occupation	Military Forces	0.016	7.33	1.45	37.07
		Unemployed and Students	0.066	4.24	0.91	19.70
		Blue Collars				
	Delusions of Infidelity (Yes)		0.003	5.23	1.77	15.49
	Visual Hallucinations (Yes)		<0.001	5.19	2.09	12.89
	Educational Level	Primary and Intermediate School	0.030	4.70	1.16	19.07
		Illiterate	0.316	4.21	0.25	69.58
		Secondary School and College				
	Associated Mood Disturbances		0.009	3.34	1.35	8.27
Delusions of Infidelity (Domain 2)	Marital Quarrels (Yes)		<0.001	7.70	2.86	20.73
	Other Substances	Alcohol	0.002	6.37	1.96	20.67
		Other Drugs	0.043	2.88	1.03	8.02
		Alone	0.007			
	Persecutory Delusions (Yes)		0.001	6.20	2.19	17.50
	Family Member Users (Yes)		0.026	3.70	1.17	11.77

When a variable has three categories, the last represents the reference category.

**Table 4 brainsci-13-01279-t004:** Summary of the logistic regression models: Goodness-of-fit, accuracy, and significance.

Outcome Variable	Model 1		Model 2		Optimal Model			
	R^2^	Statistical Accuracy	R^2^	Statistical Accuracy	R^2^	Statistical Accuracy	Sensitivity Specificity	*p*-Value
Suicidal Ideation	0.490	79.4%	0.338	76.4%	0.490	79.4%	53.8% 91.2%	0.885
Withdrawal Manifestations	0.132	68.5%	0.255	70.9%	0.255	70.9%	78.3% 57.6%	0.980
Abstinence Attempts	0.111	69.7%	0.332	75.8%	0.332	75.8%	84.3% 56.0%	0.417
Visual Hallucinations	0.667	86.1%	0.637	85.5%	0.669	87.3%	83.6% 89.8%	0.726
Auditory Hallucinations	0.561	83.6%	0.584	85.5%	0.600	85.5%	79.7% 89.6%	0.214
Persecutory Delusions	0.571	80.0%	0.533	80.0%	0.586	83.0%	85.6% 80.0%	0.627
Delusions of Infidelity	0.452	83.0%	0.419	81.2%	0.452	83.0%	65.2% 89.9%	0.059

The calculation of R^2^ represents Nagelkerke’s pseudo R^2^.

## Data Availability

All data are available upon reasonable request from the corresponding author.

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
