# Peer review of "Risk Factors of Suicidal Ideation in Iraqi Crystal Methamphetamine Users"

_brainsci, 2023, doi:10.3390/brainsci13091279_

Round 1

Reviewer 1 Report (Previous Reviewer 2)

Thank you for the opportunity to review the manuscript entitled: Risk factors of psychotic features, withdrawal manifestations, and suicidal ideation in Iraqi crystal methamphetamine users. This study presents several strenth, the authors present interesting actions for data analysis. However, several analysis require methodological arguments to better fit the aims of the study.

The authors had improved some of my previous concerns, however the aim of the study is not clear enough yet. They use parenthesis for critical factor which they are interested in studying  "(such as hallucinations and delusions)", they also mention that the oucomes are fundamental to the DSM 5 criteria, however there are no clear directions for this action in the study.

The data analysis section requires further details, for example, the authors present a hierarchical causal model in Figure 1, but they do not explain which part of the sample did they use for performing these analysis, or if they grouped theoretically or statistically.

Finally, the discussion requires to be improved in terms of formalizing the presented arguments.

Author Response

Dear Sir/Madam, Respected Reviewer,

Our detailed response to reviewer #1 is in the attached PDF file.

Thank you.

Reviewer 2 Report (New Reviewer)

Dear Authors,

I appreciated the work done by the research team.

Here are some suggestions for improving the work:

- lines 53-55 are unnecessary and can be removed;

- lines 185-187 should be included in the results at the beginning of section 3.1;

- I think it is useful to include a table to illustrate the characteristics of the patients (section 3.1)

- lines 444-449 should be included in the first part of the discussion.

Author Response

Dear Sir/Madam, Respected Reviewer,

Our detailed response to reviewer #2 is in the attached PDF file.

Thank you.

Reviewer 3 Report (New Reviewer)

The manuscript reports the findings on the correlates of psychotic features, abstinence attempts, withdrawal manifestations, and suicidal ideation in a sample of crystal methamphetamine users in Iraq. The results highlight the complex relationships with suicidal ideation in this precious sample. The authors have presented some interesting findings, which have potential clinical implications.

My main concern about this manuscript is a lack of theoretical framework in guiding the current methodology in the examination of the predictors of suicidal ideation. The study aims to “identify and analyze the risk factors associated with psychotic features (such as hallucinations and delusions), withdrawal manifestations after abstinence, and suicidal ideations in users of crystal methamphetamine in Iraq” (p. 124 – 127, p. 3), however, how these risks factors were selected, and in what ways they are clinically and theoretically meaningful, and how these relationships were conceptualized were not adequately articulated. The authors have mentioned some “etiological theories” (line 94, p. 2), but are not elaborated in detail. These theories,  which are useful to the formulation of the hypotheses and the model (ref Figure 1). should be fleshed out in the Introduction. Relatedly, the value of the “multi-domain hierarchy of causality” (line 144, p.3), although potentially promising, is not clearly explained. Importantly, how the domains of risk factors and phenomenon were defined, arranged and included are not very explicit. A lack of a rigorous theoretical framework has discredited the value of the findings.

 I also have other comments, as follows:

 Measures:

-          The authors mentioned that “the researchers developed a questionnaire designed to address the DSM-5 criteria for diagnosing stimulant use disorder. This questionnaire assessed and gathered data on the participant's substance abuse patterns, symptoms, and other relevant factors that align with the diagnostic criteria” (lines 172 – 176, p. 4). The way of validating the diagnosis of the participants (e.g. were the measures self-report or interviewer-rated?), as well as the psychometric properties of the measures of key variables (e.g. clinical manifestations such as psychotic experiences), are questionable (ref section 2.4). Please elaborate on the measures of these variables and the psychometric properties, as the credibility of the findings depends much on their measures.

-          The exact specification of the sample size calculation should be mentioned. What statistical modelling approach (e.g. regression, speculated effect sizes and power) was the calculation based on?

-          How many participants were screened and excluded (lines 187 – 190, p. 4)?

-          The study considered suicidal ideation as a binary variable, based on the cut-off of the Beck’s Suicide Ideation Scale-Current (SSI-C) (section 2.5). However, the cited evidence suggests a sensitivity which is far from satisfactory (i.e. 53%). Would the results change if suicidal ideation was treated as continuous?

-          The value of the operating characteristic curve (ROC) analysis to identify a cut-off of age for suicidal ideation (suicidal ideators vs non-ideators) is not clear. The rationale is not mentioned in the Introduction, and how the cut-off is valuable to the understanding of these risk factors is unknown.

Results:

-          Descriptive statistics reported in section 3.1 could be organized as tables for easy reference and understanding.

-          Casual language should be avoided. For example, “The multivariable analyses largely supported the pre-study hypotheses and revealed a few reciprocal causal relationships, such as between visual hallucinations and suicidal ideations” (lines 432- 434, p. 10). The same issue is also found in the Discussion.

-          The meaning of “superior outcome variables” (line 443, p. 10) is not clear. Please explain.

Discussion:

-          The Discussion is out of balance in interpreting the current findings. Some of the arguments are too far-fetched and out of context given the current results (e.g. the relevance of an accurate scale of drug use, lines 535 – 546, p. 14), while some of the arguments deserve further elaboration (e.g. military forces employees were more prone to persecutory delusions than delusions of infidelity, which explain the differential psychopathology of subtypes of delusions). A lack of a guiding framework may explain this. The Discussion requires substantial revision to highlight the value of the findings.

-          Some of the information here could be presented earlier in the Introduction to set the stage for the framework of the study, as well as the scientific value of the study. For example, the paragraph from lines 580 – 610 (p. 14-15) could be concisely put in the Introduction.

-          Only half of the participants required by the sample size calculation were recruited, which is far from satisfactory, with a high possibility that most of the analyses are underpowered. A strong justification should be made.

-          I don’t think the exclusion of participants with dual diagnoses is a limitation. Rather, it is the strength of the study to set stringent selection criteria.

-          The authors mentioned that “The scale [SSI-C]'s accuracy may also be influenced by the chosen cut-off score, which could vary across studies” (lines 648 – 649, p. 16). This is a fatal point, as the SSI-C assessed suicidal ideation is a key outcome of the study. Please rewrite.

Author Response

Dear Sir/Madam,

Kindly find the detailed reply to reviewer #3 in the attached PDF file.

Thank you.

Round 2

Reviewer 1 Report (Previous Reviewer 2)

Thank you for attending my comments.

Author Response

Dear Sir/Madam, the Respected Reviewer,

We sincerely thank you for your thoughtful expressions and guidance during the peer review. Your contributions have served as a source of inspiration and have significantly contributed to enhancing the scholarly quality of our article, elevating it to its present state.

We wish to reiterate our heartfelt thanks for your dedicated efforts.

Best regards,

Ahmed.

Reviewer 3 Report (New Reviewer)

Thank you so much for the revision and responses to my comments. As noted in my previous review, the manuscript reports the complex relationship of psychotic features, abstinence attempts, withdrawal manifestations with suicidal ideation in a sample of crystal methamphetamine users in Iraq. The authors are commended for providing a thorough response to my suggestions and a clear summary of the revisions. In particular, I appreciate very much the detailed elaboration and justification of the design of the study, measures and analysis, as well as the authors’ effort to make a balance among views from several reviewers to come up with the current version of the manuscript. The revised manuscript is now self-explanatory in content and coherent to read, with sufficient details for the theoretical background and analysis. My final suggestion is to include (brief) explanations/ speculations on the relationship between a lack of trauma and suicidal ideation (OR=4.58, p=0.006), which seems counter-intuitive to our understanding for trauma as a risk factor to various psychopathology (including suicidality).  

Author Response

Dear Sir/Madam, Respected Peer-Reviewer,

Kindly find our reply in the attached PDF file. Once again, thank you for your time and efforts.

Best regards,

Ahmed.

This manuscript is a resubmission of an earlier submission. The following is a list of the peer review reports and author responses from that submission.

Round 1

Reviewer 1 Report

This study aims to investigate the risk factors of psychotic features (hallucinations and delusions), substance abstinence (cessation) attempts, withdrawal manifestations following abstinence, and suicidal ideations in 165 Iraqi crystal methamphetamine users.

There are several weaknesses in the study:

- the a priori power analysis showed that a sample size of 384 patients was needed, but authors were able to recruit only 165: this is half the sample size and although the post hoc power analysis showed some adequacy, this is undoubtedly a limitation.

- there is no mention of correction multiple testing 

- ROC curves are performed for males and females but the N of women is really small and I wonder how PPV and NPV are…

- in general the hypotheses are stated in an unclear manner and the presentation of results does not really focus on a specific hypothesis

- the authors should simplify the paper and probably focus on only one-two main research questions, this seems more of an adaptation from a PhD thesis and is too digressive

Language is ok but the text needs to be streamlined

Reviewer 2 Report

It is an interesting work with a brilliant introduction, it presents some methodological issues and some mild correction needs in the results section. However, I found some sensitive information in the discussion section, possibly misunderstanding topics not directly related to the variables of interest, and out of the scope of the Journal. All my comments and concerns are written in margins of the attached document.
